# Decreased Levels of GSH Are Associated with Platinum Resistance in High-Grade Serous Ovarian Cancer

**DOI:** 10.3390/antiox11081544

**Published:** 2022-08-10

**Authors:** Daniela Criscuolo, Rosario Avolio, Matteo Parri, Simona Romano, Paola Chiarugi, Danilo Swann Matassa, Franca Esposito

**Affiliations:** 1Department of Molecular Medicine and Medical Biotechnology, University of Naples Federico II, 80131 Naples, Italy; 2Department of Experimental and Clinical Biomedical Sciences, University of Florence, 50134 Florence, Italy

**Keywords:** ovarian cancer, platinum resistance, GSH, reactive oxygen species (ROS)

## Abstract

High-grade serous ovarian cancer (HGSOC) is the most common and aggressive OC histotype. Although initially sensitive to standard platinum-based chemotherapy, most HGSOC patients relapse and become chemoresistant. We have previously demonstrated that platinum resistance is driven by a metabolic shift toward oxidative phosphorylation via activation of an inflammatory response, accompanied by reduced cholesterol biosynthesis and increased uptake of exogenous cholesterol. To better understand metabolic remodeling in OC, herein we performed an untargeted metabolomic analysis, which surprisingly showed decreased reduced glutathione (GSH) levels in resistant cells. Accordingly, we found reduced levels of enzymes involved in GSH synthesis and recycling, and compensatory increased expression of thioredoxin reductase. Cisplatin treatment caused an increase of reduced GSH, possibly due to direct binding hindering its oxidation, and consequent accumulation of reactive oxygen species. Notably, expression of the cysteine-glutamate antiporter xCT, which is crucial for GSH synthesis, directly correlates with post-progression survival of HGSOC patients, and is significantly reduced in patients not responding to platinum-based therapy. Overall, our data suggest that cisplatin treatment could positively select cancer cells which are independent from GSH for the maintenance of redox balance, and thus less sensitive to cisplatin-induced oxidative stress, opening new scenarios for the GSH pathway as a therapeutic target in HGSOC.

## 1. Introduction

Ovarian cancer (OC) is the most aggressive gynecological malignancy and the fifth leading cause of death among women [1]. According to the latest Global Cancer Observatory (GLOBOCAN) report, OC accounted for 1.6% of all cancers and 2.1% of all cancer deaths worldwide in 2020 [2]. OC is a heterogenous disease which encompasses a group of neoplasms with distinct cells of origin, risk factors, clinical and molecular features, and prognosis. The most common subtypes, epithelial OCs (EOCs) account for 90% of ovarian tumors. The predominant subtype, the high-grade serous ovarian cancer (HGSOC)—which accounts for 70–80% of all EOC subtypes [1]—is commonly diagnosed at advanced stages (III or IV) when the disease is disseminated [3]. Primary debulking surgery and adjuvant platinum-based chemotherapy are the standard of care for HGSOC patients [4]. Although the initial response rates to systemic chemotherapy exceed 80% when integrated with primary cytoreductive surgery, most patients still develop a recurrent disease [5], with over 50% of patients eventually dying within 5 years of diagnosis [6]. The major cause of cancer recurrence is resistance to platinum-based agents, that can be “intrinsic” in 10% to 15% of HGSOC patients, defined as platinum-refractory, or developed within 6 months of completing chemotherapy for approximately 20% to 30% of patients, named platinum-resistant [5]. Platinum resistance is a complex and multifactorial phenomenon involving alterations in drug influx and/or efflux, DNA damage repair pathways, and cell death signaling [7].

The antitumor activity of cisplatin was initially attributed to its ability to bind nuclear DNA, generating protein—DNA complexes and DNA adducts, which interfere with transcription and replication processes leading to massive cell death [8]. Nowadays, this view appears as an oversimplification, considering that: (i) only ~1% of intracellular cisplatin interacts with nuclear DNA, and (ii) cisplatin have cytotoxic effects even in enucleated cells [8,9]. It is now clear that cisplatin exerts a cytoplasmic toxicity by binding various intracellular molecules, with reduced glutathione (GSH) being its main target [8,10,11].

GSH represent the most important antioxidant defense against free radicals, and a useful metabolite in multiple processes, including cellular differentiation and proliferation [12]. This tripeptide is synthesized in the cytosol and distributed to almost all organelles where it mainly exists in its reduced form, except for the endoplasmic reticulum, which needs to maintain a highly oxidized environment and thus contains high levels of glutathione oxidized species (GSSG) [13]. The binding of cisplatin to GSH alters the redox balance leading to accumulation of reactive oxygen species (ROS), which can directly induce apoptosis by opening the mitochondrial permeability transition pore [11]. A fine-tuned balance between antioxidant defenses and ROS levels is essential to ensure physiological processes and cell survival. Indeed, cancer cells, which display increased ROS generation due to mitochondrial dysfunction, altered metabolism, and frequent genetic mutations, concomitantly enhance the levels of antioxidants to keep this balance [14]. While moderate levels of ROS can promote tumorigenesis by increasing cell proliferation, survival, and mutation rate, and sustain tumor progression by promoting cancer cell motility [15], when this balance is disrupted, the levels of ROS overcome the antioxidant defenses, causing detrimental oxidative stress that damages macromolecules and leads to cell death [16]. In line with this evidence, the use of natural as well as synthetic compounds able to increase ROS levels above critical levels are emerging as possible therapeutic strategies to treat cancer [17,18]. The transcription factor NRF2 (nuclear factor erythroid 2-related factor 2) plays a crucial role in promoting oxidative stress tolerance by regulating the expression of several antioxidant genes, including enzymes involved in glutathione synthesis and recycling (GCGL, GCLM, GSR) and in the thioredoxin system (TXN, TXNRD1, TXNIP) [19]. The NRF2 pathway, neutralizing cellular ROS, protects cancer cells from ROS-mediated cytotoxicity induced by chemotherapeutic drugs [20]. Consistently, NRF2 activation has been reported to promote platinum resistance in OC [21]. Moreover, NRF2 rewires cellular metabolism to improve antioxidant response by promoting the metabolic pathways that support antioxidant systems and repair ROS-induced damage (i.e., the pentose phosphate pathway, metabolism of TCA cycle intermediates, the nucleotide biosynthesis pathway, and the glucuronidation pathway), and by repressing NADPH-consuming processes, such as lipid biosynthesis, to spare essential cofactors for detoxification reactions [19].

Since energy metabolism and ROS homeostasis reciprocally influence each other [22], a better understanding of the interplay between metabolic plasticity and chemotherapy-induced oxidative stress will help to identify new pharmacological targets to overcome chemoresistance. Indeed, metabolic reprogramming strongly contributes to chemoresistance in several tumor contexts [23]. In particular, drug resistant OC cells often show a highly active metabolic state, that can be induced by cisplatin exposure, which allows them to switch between glycolysis and oxidative phosphorylation (OXPHOS) to produce energy, increase tumor growth, and survive chemotherapy [24]. Furthermore, metastatic OC cells in the omentum, growing in close proximity with adipocytes, enhance fatty acid uptake, by upregulating fatty acid receptor CD36, and utilization via β-oxidation, by upregulating the expression of CPT1A, leading to an increase in OXPHOS activity [25,26]. Notably, inhibition of both OXPHOS and β-oxidation improves the sensitivity of OC cells to platinum compounds [27,28]. In line with this evidence, we have previously demonstrated that drug resistant OC cells rely on OXPHOS for their energetic metabolism and that increased mitochondrial respiration is responsible for the chemoresistant phenotype by promoting the activation of a pro-inflammatory program [27]. Furthermore, we have shown that drug resistant OC cells remodel their cholesterol metabolism by reducing the biosynthetic pathway and increasing exogenous cholesterol uptake through the low-density lipoprotein (LDL) receptor, in order to acquire chemoresistance [29].

Herein, we show that cisplatin-resistance is associated with alterations in the antioxidant network in HGSOC cells. Starting from a metabolomic analysis, we show that cisplatin-resistant cells display decreased levels of reduced glutathione, and lower expression levels of enzymes involved in GSH biosynthesis and recycling. Interestingly, cisplatin treatment leads to a significant increase in both cytoplasmic and mitochondrial ROS only in platinum sensitive cells, rendering them more susceptible to oxidative stress-induced death. Accordingly, low expression of some of the enzymes involved in GSH synthesis and trafficking (xCT and SLC25A11) correlates with bad post-progression survival (PPS) and poor response to therapy in HGSOC patients, whereas thioredoxin reductase (TRXRD1) expression has an inverse correlation. Our data support a model in which cisplatin treatment positively selects cancer cells that do not rely on GSH to maintain redox balance in order to prevent cisplatin-induced oxidative stress.

## 2. Materials and Methods

### 2.1. Metabolomic Analysis by Gas Chromatography-MS (GC-MS)

For SCAN mode, 10^6^ cells were collected and subjected to extraction using a mixture of CHCl_3_:Methanol:H_2_O (#34854-1 and #900688-1, Sigma-Aldrich, St. Louis, MO, USA; #102699-1000, Merck, Burlington, MA, USA) (1:1:1). Cells were quenched with 0.4 mL ice cold methanol and an equal volume of water containing 1 μg norvaline (#53721, Sigma-Aldrich, St. Louis, MO, USA), used as internal standard. One volume of chloroform was added, and the samples were vortexed at 4 °C for 30 min. Samples were centrifuged at 3000× *g* for 10 min, and the aqueous phase was collected in a new tube and evaporated at room temperature. Dried polar metabolites were dissolved in 60 μL of 2% methoxyamine hydrochloride (#226904, Sigma-Aldrich, St. Louis, MO, USA) in pyridine (25104, Thermo Fisher Scientific Inc., Waltham, MA, USA) and held at 30 °C for 2 h. After dissolution and reaction, 90 μL MSTFA + 1% TMCS (69478-10x, Sigma-Aldrich, St. Louis, MO, USA) were added, and samples were incubated at 37 °C for 60 min. Gas chromatographic runs were performed with Agilent GC-Intuvo9000\MS-5977B with helium as carrier gas at 0.6 mL/min. The split inlet temperature was set to 250 °C and the injection volume was 1 μL. A split ratio of 1:10 was used. The GC oven temperature ramp was from 60 °C to 325 °C at 10 °C/min. The data acquisition rate was 10 Hz. For the quadrupole, an EI source (70 eV) was used, and full-scan spectra (mass range from 50 to 600) were recorded in the positive ion mode. The ion source and transfer line temperatures were set, respectively, to 250 °C and 290 °C. The MassHunter data processing tool (Agilent, Santa Clara, CA, USA) was used to obtain a global metabolic profiling, together with the Fihen Metabolomics RTL library (G1676AA, Agilent, Santa Clara, CA, USA). For determination of relative metabolite abundances, the integrated signal of all ions for each metabolite fragment was normalized by the signal from norvaline and the per cell number.

### 2.2. Cell Culture, Cell Treatments and Transfections

The paired HGSOC cell lines PEA1/PEA2, PEO14/PEO23 and PEO1/PEO4 have been described elsewhere [30]. All lines were maintained in RPMI 1640 media with 10% fetal bovine serum, glutamine, and Normocin (Invivogen, San Diego, CA, USA), at 37 °C, 5% CO_2_. In glutathione colorimetric detection assay, PEA1 and PEA2 cells were treated with 20 μM and 40 μM of cisplatin, respectively, for 30 min. To evaluate intracellular ROS levels, as well as for real-time PCR assays, PEA1 and PEA2 cells were treated with 10–20 μM and 20–40 μM of cisplatin, corresponding to the IC50 of each cell line (20 and 40 µM, respectively) and the relative half, for 24 h. Cisplatin (NSC 119875) was purchased from Selleckchem (Cat. Number S1166). Transient transfections of siRNAs were performed using Lipofectamine 3000 Reagent (L3000001, Thermo Fisher Scientific Inc., Waltham, MA, USA) according to the manufacturer’s protocol. TRXNRD1-directed siRNAs were purchased from Thermo Fisher Scientific Inc., Waltham, MA, USA (Silencer Select siRNA, catalog # 4390824), along with the negative control siRNA (catalog # 4390843).

### 2.3. Glutathione Colorimetric Detection Assay

A glutathione colorimetric detection kit (#EIAGSHC, Thermo Fisher Scientific Inc., Waltham, MA, USA) was used to determine the reduced (GSH) and oxidized (GSSG) glutathione levels according to the manufacturer’s instructions. Briefly, cell pellets were washed in ice-cold PBS and resuspended in ice-cold 5% salicylic acid (SSA), followed by centrifugation at 14,000 rpm for 10 min at 4 °C. The resulting supernatant was aliquoted for measurement of total glutathione (GSH and GSSG) and GSSG. For GSSG measurements, the samples were treated with 2-vinylpyridine (2VP) solution for 1 h at room temperature. The standards and sample dilutions and assay were conducted according to the manufacturer’s instructions. Finally, the plate was incubated for 20 min at room temperature before reading the absorbance at 405 nm in a microplate reader (Synergy H1 from BioTek, Winooski, VT, USA). Free GSH was determined by subtracting the GSSG concentration from the total GSH.

### 2.4. RNA Extraction and Real-Time Reverse Transcriptase-Polymerase Chain Reaction (RT-PCR)

Total RNA extraction procedures were performed by using TRI Reagent (T9424, Sigma-Aldrich, St. Louis, MO, USA), following the manufacturer’s instructions. For first-strand synthesis of cDNA, 1 μg of RNA was used in a 20 μL reaction mixture by using a SensiFast cDNA synthesis kit (BIO-65054, Meridian Bioscence, Cincinnati, OH, USA). For real-time PCR analysis, 0.4 μL of cDNA sample was amplified by using the SensiFast Syber (BIO-98050, Meridian Bioscence, Cincinnati, OH, USA) in an iCycler iQ Real-Time Detection System (Bio-Rad Laboratories GmbH, Segrate, Italy). The reaction conditions were 95 °C for 2 min, followed by 40 cycles of 5 s at 95 °C and 30 s at 60 °C. Cyclophilin A (PPIA) was chosen as the internal control. The primers used for PCR analysis are listed in Table 1.

### 2.5. Western Blotting Analyses

Equal amounts of protein from cell lysates were subjected to SDS-PAGE and transferred to a PVDF membrane (Millipore, Burlington, MA, USA). Membranes were blocked in a 5% milk TBS-tween solution for 1 h at room temperature. The following antibodies were used: anti-β-actin (sc-69879, Santa Cruz Biotechnology, Dallas, TX, USA), anti-F1ATPase (sc-58619, Santa Cruz Biotechnology, Dallas, TX, USA), anti-GAPDH (sc-69778, Santa Cruz Biotechnology, Dallas, TX, USA), anti-GCLC (sc-390811, Santa Cruz Biotechnology, Dallas, TX, USA), anti-HSP60 (sc-1052, Santa Cruz Biotechnology, Dallas, TX, USA), anti-TXNRD1 (sc-28321, Santa Cruz Biotechnology, Dallas, TX, USA), anti-goat (sc-2020, Santa Cruz Biotechnology, Dallas, TX, USA), anti-mouse (A90-137P, Bethyl Laboratories, Montgomery, TX, USA), anti-rabbit (A120-101P, Bethyl Laboratories, Montgomery, TX, USA). All primary antibodies were used at a 1:1000 dilution and incubated O/N at 4 °C. Anti-goat antibody was used at a 1:4000 dilution and incubated for 1 h at room temperature. Anti-mouse and anti-rabbit antibodies were used at a 1:20,000 dilution and incubated for 1 h at room temperature. Images were acquired using the ChemiDoc MP system (Bio-Rad Laboratories GmbH, Segrate, Italy). Quantitative estimation was performed by measuring densitometric band intensity using ImageJ [31].

### 2.6. Assessment of Intracellular ROS Levels

PEA1 and PEA2 cells were plated in 35-mm dishes and cultured overnight. Then the medium was replaced with cisplatin. After 24 h, cells were washed with PBS, trypsinized, and collected. Intracellular ROS levels were assessed by incubating cells with H2DCFDA (1 µM; Thermo Fisher Scientific Inc., Waltham, MA, USA) or MitoSOX (5 µM; Thermo Fisher Scientific Inc., Waltham, MA, USA) at 37 °C for 10 min and 30 min, respectively. Mean fluorescence intensity (MFI) of 10,000 cells was analyzed in each sample by using a MACSQuant Analyzer 10 flow cytometer (Miltenyi Biotec, Bergisch Gladbach, Germany) or a BD AccuriTM C6 Cytometer (BD Biosciences, Franklin Lakes, NJ, USA). The results were analyzed using FlowJo™ v10.8 Software (BD Biosciences, Franklin Lakes, NJ, USA).

### 2.7. Viability Assays

In cell viability experiments, cells were plated in monolayer in complete medium and TXNRD1 silencing was induced for 24 h. Following TXNRD1 silencing, cisplatin was added to the medium (20 µm for PEA1 and 40 µm for PEA2 cells) for 48 h, while H_2_O_2_ (20 mM) was added to the media for 30 min. Cell viability was measured by MTT assay by using the in vitro toxicology assay kit (TOX1-1KT, Sigma-Aldrich, St. Louis, MO, USA), following the manufacturer’s instructions.

### 2.8. Statistical Analyses

Statistical analyses were performed by using GraphPad Prism software, version 8 (San Diego, CA, USA). The two-tailed unpaired Student’s *t*-test was used to establish the statistical significance of changes in metabolite levels in untargeted metabolomic analysis, changes in gene expression levels compared to controls in qPCR experiments, and changes of densitometric band intensity in Western blots. A ratio paired Student’s *t*-test was used to establish the statistical significance of changes in glutathione assay. *p* < 0.05 was considered statistically significant.

### 2.9. Estimates of Survival and Response to Platinum-Based Therapy

Correlation between expression levels of genes involved in antioxidant defence and response to platinum-based therapy in HGSOC patients was investigated by using an ROC plotter [32]. Correlation between expression levels of genes involved in antioxidant defence and post-progression survival in HGSOC patients was investigated by using a Kaplan–Meier plotter [33].

## 3. Results

### 3.1. Reduced Glutathione Levels Are Decreased in Ovarian Cancer Cells

Previous analyses performed in two HGSOC cell lines (PEA1/PEA2), isolated from the same patient before chemotherapy and following the onset of acquired clinical platinum resistance, respectively, allowed us to identify remodeling of energy metabolism processes as being largely responsible for the acquisition of drug resistance [27,30]. In order to obtain a broader picture of the metabolic profile of platinum-resistant OC cells, we performed an untargeted metabolomics analysis by evaluating the levels of all measurable analytes in PEA1 and PEA2 cells. Surprisingly, this analysis showed a significant reduction of analytes implicated in antioxidant defence, such as GSH, hypotaurine, and taurine in cisplatin-resistant PEA2 cells (Figure 1A). To support this result, GSH levels were measured under basal conditions in PEA1 and PEA2 cells and in two additional matched pairs of cisplatin sensitive and resistant cell lines (namely, PEO1/PEO4 and PEO14/PEO23). We found that GSH levels are dramatically reduced in PEA2 cells compared to PEA1 cells, confirming the metabolomics data; accordingly, PEO23 also showed a significant, although less striking, decrease of GSH levels (Figure 1B). Although the PEO1/PEO4 couple did not reproduce the same results, it is worth noting that PEO1 cells, as opposed to PEA1 and PEO14, are not chemo-naïve, as they are derived from a patient at first relapse after cisplatin-based chemotherapy [30], although still clinically responsive to the treatment, and indeed show the lowest levels of GSH among the cisplatin-sensitive cell lines. Of note, even in this couple, chemoresistant cells do not show increased GSH levels despite their increased oxidative metabolism. Indeed, we calculated the bioenergetic cellular (BEC) index, a measure of cellular metabolic state defined by the ratio between the protein levels of F1ATPase, HSP60 and GAPDH (Figure 1D). This procedure reflects results from other metabolism-related assays such as the Seahorse, as elsewhere described [27,34]. The BEC index can be considered a first choice method when metabolism assessment has to be performed on tissues and biopsy specimens [35]. Considering that cancer cells relying on oxidative metabolism for their bioenergetic needs commonly show upregulation of ROS buffering systems to counteract oxidative stress generated by dysregulated metabolism, and to explain these unexpected results, we hypothesized that a decrease of GSH in chemoresistant cells could result from cisplatin binding to GSH, as reported in previous studies [11], thus exerting selective pressure against GSH dependence for ROS detoxification [11]. To explore this hypothesis, we analyzed glutathione (GSH and GSSG) levels in PEA1 and PEA2 cells after cisplatin treatment (Figure 1C). The GSH assay showed increased levels of GSH in both cell lines following cisplatin exposure, but only in PEA1 cells was this increase statistically significant. Considering that this phenomenon is observed at very early treatment times, this could point toward the formation of intracellular GSH-cisplatin adducts, which prevented GSH oxidation, although further studies are needed to directly demonstrate it. Taken together, these data suggest that addiction to GSH in a highly oxidative environment would be counter-selected by cisplatin treatment and that the development of alternative antioxidant pathways could therefore favor the selection of cisplatin-resistant clones.

### 3.2. Cisplatin Treatment Triggers ROS Generation

GSH represents one of the major antioxidant defenses against free radicals. It is synthesized exclusively in the cytosol, but it is then distributed to different organelles, including mitochondria, which represent the major source of ROS originating from the electron transport chain [36]. When GSH is unavailable, ROS cannot be neutralized and promotes oxidative damage [37]. To further support the hypothesis that cisplatin treatment causes oxidative stress by hampering the GSH pathway, we evaluated the levels of intracellular ROS upon cisplatin treatment in PEA1 and PEA2 cells using two different drug concentrations corresponding to the IC_50_ of each cell line (20 and 40 µM, respectively) and the relative half (10 and 20 µM, respectively). To this purpose, we loaded cells with 2′,7′-dichlorofluorescein diacetate, a ROS-sensitive fluorescent probe, and measured cytoplasmic ROS levels by flow cytometry. As shown in Figure 2A, cisplatin treatment is able to increase cytoplasmic ROS levels both in PEA1 and PEA2 cells. Nevertheless, PEA2 cells need a doubled amount of cisplatin to reach the same ROS increase observed in PEA1 cells (Figure 2A). We also evaluated the levels of mitochondrial ROS by flow cytometry using the fluorescent probe MitoSOX Red. Results showed a significant increase in mitochondrial ROS following cisplatin treatment only in sensitive PEA1 cells (Figure 2B). Overall, these results suggest that platinum resistant OC cells might rely more on alternative antioxidant mitochondrial defenses rather than GSH, whose levels are decreased, to ensure a balanced redox homeostasis, thus preventing the cytotoxic effects of cisplatin on the GSH-dependent ROS detoxification pathway.

### 3.3. GSH Pathways Are Dysregulated in Chemoresistant Ovarian Cancer Cells

To further investigate the regulation of the GSH pathway in resistant HGSOC cells, we evaluated the expression levels of genes involved in GSH biosynthesis and recycling, and alternative antioxidant systems, in all three matched pairs of resistant OC cell lines compared to their sensitive counterparts. GSH is produced by de novo biosynthesis involving a two-step reaction catalyzed by gluthatione synthase (GSS) and a glutamate–cysteine ligase catalytic subunit (GCLC) [38], or it can be recycled from GSSG by glutathione disulfide reductase (GSR) [39] (Figure 3A). We also analysed the expression of the sodium-independent cystine–glutamate antiporter (xCT or SLC7A11), which participates in the biosynthetic pathway primarily importing cystine, the oxidized dimeric form of cysteine, glutathione peroxidase 4 (GPX4), which can catalyze the reduction of peroxides at the expense of glutathione, and glutathione-specific gamma-glutamylcyclotransferase 1 (CHAC1), which decreases intracellular GSH by digesting glutathione into 5-oxoproline and Cys-Gly dipeptide. As a potential compensatory pathway, we evaluated the expression of thioredoxin reductase 1 (TXNRD1), a gene encoded for an essential antioxidant enzyme that reduces thioredoxin, as well as other compounds, that in turn reduce several target proteins from their oxidized forms [40]. Analyses showed that all platinum resistant cells, except PEO4, exhibit reduced or unchanged levels of the transcripts coding for proteins involved in GSH homeostasis, and that, in parallel, TXNRD1 levels are increased in both PEA2 and PEO4 cells (Figure 3B). GPX4 and CHAC1 levels were unchanged in PEA2 cells, but, respectively, lower and higher in both PEO4 and PEO23 cells compared to their sensitive counterparts. Furthermore, Western blot analyses showed that, although the mRNA levels were unchanged, GCLC protein levels were increased in PEA2 and PEO23 cells compared to their matched sensitive cells, while TRXNRD1 protein is also upregulated in PEO23 (Figure 3C). These data are in support of reduced GSH biosynthesis and recycling in resistant OC cells, which rather rely on alternative antioxidant defense mechanisms. To test this latter hypothesis, we compared PEA1 and PEA2 cell viability following cisplatin treatment upon TXNRD1 silencing. Moreover, to support the essential role of TXNRD1 as an antioxidant system in PEA2 cells, we induced oxidative stress with H_2_O_2_. Strikingly our results showed that TXNRD1 silencing sensitizes to both cisplatin- and H_2_O_2_-induced cell death in cisplatin-resistant PEA2 cells only (Figure 3D). Of note, it has been previously shown that auranofin, a thioredoxin reductase inhibitor, is effective in inducing apoptosis in cisplatin-resistant human ovarian cancer cells [41].

### 3.4. Dysregulated Expression of Antioxidant Enzymes Correlate with Response to Therapy and Survival in HGSOC Patients

Resistance to platinum-based agents is the major cause of HGSOC recurrence; therefore, there is an urgent need to identify alternative therapeutic strategies to improve patient survival [42]. To further investigate the role of antioxidant defenses in this context, and their potentiality as predictive biomarkers, we interrogated the ROC plotter database [32], which enables the search for correlations between gene expression and response to therapy using transcriptome-level data. Results showed that SLC7A11 (xCT) is expressed at a significantly lower level (AUC: 0.566—ROC *p* = 0.0037, Mann–Whitney test *p* = 0.0074, chi-squared *p* = 0.0536) in patients who do not respond to platinum-based therapy (non-responders) at 12 months of follow-up (Figure 4A). GPX4 showed the same trend (Appendix A), in line with a correlation between reduced GSH and chemoresistance. Next, we interrogated the Kaplan–Meier plotter database [33] to look for the impact of GSH-related enzymes on the outcome of HGSOC and, in particular, their correlation with post-progression survival (PPS), defined as the time interval from the date of recurrence to the date of death, following therapy containing platin, as a better measure of acquired chemoresistance. Strikingly, expression of SLC7A11 significantly correlated with PPS (*p* = 0.00059): patients with high SLC7A11 (above median) expression have a median survival of almost 45 months, while those with low (below median) expression of SLC7A11 show a decrease in median survival to 35 months (Figure 4B). We obtained similar results for GPX4 and GPX1 (Appendix A), and SLC25A11, predominantly responsible for GSH import into mitochondria (Figure 4C), in line with the hypothesis of limited reliance on the GSH pathway and activation of alternative antioxidant pathways in resistant cells. Accordingly, when we searched for the correlation between TXNRD1 and PPS, we found an opposite correlation: patients with high TXNRD1 expression levels had worse PPS (*p* = 0.026), with a median survival rate of 38 months against the 45 months of those showing low levels of this gene (Figure 4C). Altogether, these data support the hypothesis that remodeling of redox metabolism could contribute to acquisition of platinum resistance in HGSOC.

## 4. Discussion

EOC is the most common gynecological malignancy, accounting for 90% of OCs. Among all subtypes, HGSOC is responsible for the majority of OC deaths, mainly due to its aggressive behavior and late-stage diagnosis [3]. Currently, the standard first-line treatment is mainly based on a combination of platinum compounds and taxanes. Despite a good initial response, most patients with HGSOC frequently relapse and become resistant to chemotherapy [43]. In recent decades, several efforts have been directed to improve progression-free survival time, revert drug resistance, or develop new therapeutic approaches for advanced HGSOCs. However, the development of platinum resistance remains a major clinical challenge. Consequently, a better understanding of the mechanisms underlying platinum resistance in HGSOC is urgently needed. In this context, recent studies highlighted the role of cancer metabolism in the development of chemoresistance [44]. Indeed, considerable evidence has showed that both glycolysis and mitochondrial metabolism are critical for cancer progression [45]. Furthermore, it has been demonstrated that tumor progression, as well as drug resistance, are frequently associated with increased reliance on oxidative metabolism in several tumor types [46,47,48,49]. We have previously demonstrated that platinum-resistant HGSOC cells show increased dependence on OXPHOS [27], which leads to drug resistance through the activation of an inflammatory response [50], accompanied by reduced cholesterol biosynthesis and increased uptake of exogenous cholesterol via the LDL receptor [29].

Here, we have added a new component to the intricate metabolic remodeling underlying development of platinum resistance in HGSOCs. Taking advantage of an untargeted metabolomics analysis, we found that resistant cells display reduced GSH levels despite their increased oxygen consumption rate. Commonly, it is expected that cancer cells with increased OXPHOS exhibit a parallel increase in antioxidant defenses to counteract the ROS frequently generated by altered metabolism. However, of note, we found that this was not the case in our experimental system. In particular, PEA2 and PEO23 drug-resistant cells showed a significant reduction of GSH levels compared to their sensitive counterparts, albeit at different degrees, while only the PEO1/PEO4 couple showed no difference, but also the lowest GSH levels among all matched-pair cell lines. In this regard, it is worth noting that PEA1 and PEO14 cells were established from chemotherapy naïve HGSOC patients, and their counterparts PEA2 and PEO23 were isolated upon relapse with drug resistant disease 5 and 7 months after chemotherapy, respectively [30]; on the other hand, PEO1 and PEO4 cells were established after many months of therapy, as they originated from the first chemo-responsive relapse at 22 months of treatment (PEO1), and from a second drug-resistant relapse occurring after a further 10 months of therapy (PEO4) [30]. This evidence suggests that, in the highly oxidative OC environment, cisplatin treatment could favor the selection of cells with low GSH levels by killing those mainly relying on GSH for redox balance, through direct binding and impairment of its function.

However, the role of GSH in the development of chemoresistance is controversial. Indeed, GSH can contribute to chemoresistance since the formation of adducts with cisplatin results in a reduced concentration of active drug within the cells; whereas, on the other hand, cisplatin-GSH adducts also lower the intracellular GSH levels, limiting the antioxidant capacity of cancer cells, thus contributing to cisplatin cytotoxicity [51]. Our data could be interpreted in light of the second hypothesis, since short exposure to cisplatin increased the intracellular levels of GSH mainly in sensitive cells, which could be due to accumulation of cisplatin-GSH adducts. In line with an imbalanced redox homeostasis consequent to the formation of cisplatin-GSH adducts, cisplatin treatment led to increased cytoplasmic ROS both in sensitive and resistant cells (Figure 5). Nevertheless, resistant cells needed to be treated with a doubled amount of drug to reach the same ROS increase observed in their sensitive counterparts, most likely because of reduced formation of cisplatin-GSH adducts due to decreased GSH levels. In this regard, although the observed phenomenon could also be explained by reduced cisplatin uptake and/or increased cisplatin efflux, it is worth noting that platinum-resistant PEA2, PEO4 and PEO23 cells do not show significantly lower levels of intracellular cisplatin in comparison to their matched sensitive PEA1, PEO1 and PEO14 cells [52]. Moreover, measurement of mitochondrial ROS upon cisplatin treatment did not show any significant increase in resistant cells, suggesting that these cells could activate alternative antioxidants defenses independent from GSH. Of note, expression of SLC25A11, predominantly responsible for GSH import into mitochondria, directly correlates with PPS in HGSOC patients. Accordingly, all resistant cells analyzed, except PEA2, showed reduced expression of SLC7A11, which is crucial for GSH biosynthesis, and a compensatory increase in the expression of the antioxidant molecule TXNRD1; these two genes display opposite correlation with PPS, with low expression of SLC7A11, and high expression of TXNRD1 correlated with a worse PPS in HGSOC patients. Moreover, SLC7A11 expression is significantly lower in patients not responding to platinum-based therapy, suggesting that it could be worth exploring the potentiality of this gene as a predictive biomarker. Therefore, TXNRD1 increased levels could represent a way adopted by chemoresistant cells to avoid “addiction” to GSH, and thus overcome the cytotoxicity derived from cisplatin-induced reduction of GSH availability (Figure 5). Indeed, when redox balance is disrupted as a consequence of GSH depletion, ROS overcome their physiological levels and trigger oxidative damage, ultimately leading to programmed cell death [53]. Our data support the hypothesis that while sensitive HGSOC cells are mainly targeted to oxidative-induced cell death as a consequence of cisplatin-induced oxidative stress, the resistant cells, which rely on alternative antioxidants and have lowered their GSH levels, could be mainly targeted to other forms of cell death. These data are in agreement with other studies reported in the literature in which targeting antioxidant enzymes resulted in increased sensitivity of cancer cells to cisplatin [54]. Indeed, suppressing the antioxidant defenses is considered a promising strategy for cancer treatment [55].

Overall, our data suggest that cisplatin treatment could positively select cancer cells which are independent from GSH for the maintenance of redox balance, and thus less sensitive to cisplatin-induced oxidative stress. This would be particularly relevant in OC, since it has been extensively demonstrated that OXPHOS plays a relevant role in this tumor type compared to most cancers [50]. Accordingly, a recent multi-omic profiling of the same platinum-resistant and sensitive OC cell models used in the present study highlighted that OXPHOS and fatty acid oxidation are implicated in platinum resistance [28]. In recent years, evidence has been accumulating on the relevance of lipid metabolism in disease progression and acquisition of chemoresistance in OC [29,56,57]. It is worth noting that both lipid biosynthetic pathways and antioxidant defenses require a high amount of NADPH as a cofactor, potentially leading to competition, and/or to high dependence from NADPH generating/regenerating pathways, such as fatty acid oxidation itself, the pentose phosphate pathway, and activity of malic enzyme and isocitrate dehydrogenase 1 and 2. Moreover, metabolic intermediate of the cholesterol synthesis, such as squalene, can play key functions in preventing oxidative cell death [58]; as a consequence, cholesterol auxotrophy, which allows squalene accumulation, could provide a survival advantage to tumor cells under conditions of oxidative stress. This model would also correlate our previous observations showing increased reliance of chemoresistant cells on exogenous cholesterol [29] to the data herein presented, suggesting oxidative cell death as the main cisplatin mode of action in HGSOC.

These hypotheses deserve deeper investigations, as they hold potential for novel combinatorial therapeutic options, i.e., compounds that reduce the ability of cancer cells to produce NADPH coupled to lipid deprivation, which would be particularly toxic in this setting.

## Figures and Tables

**Figure 1 antioxidants-11-01544-f001:**
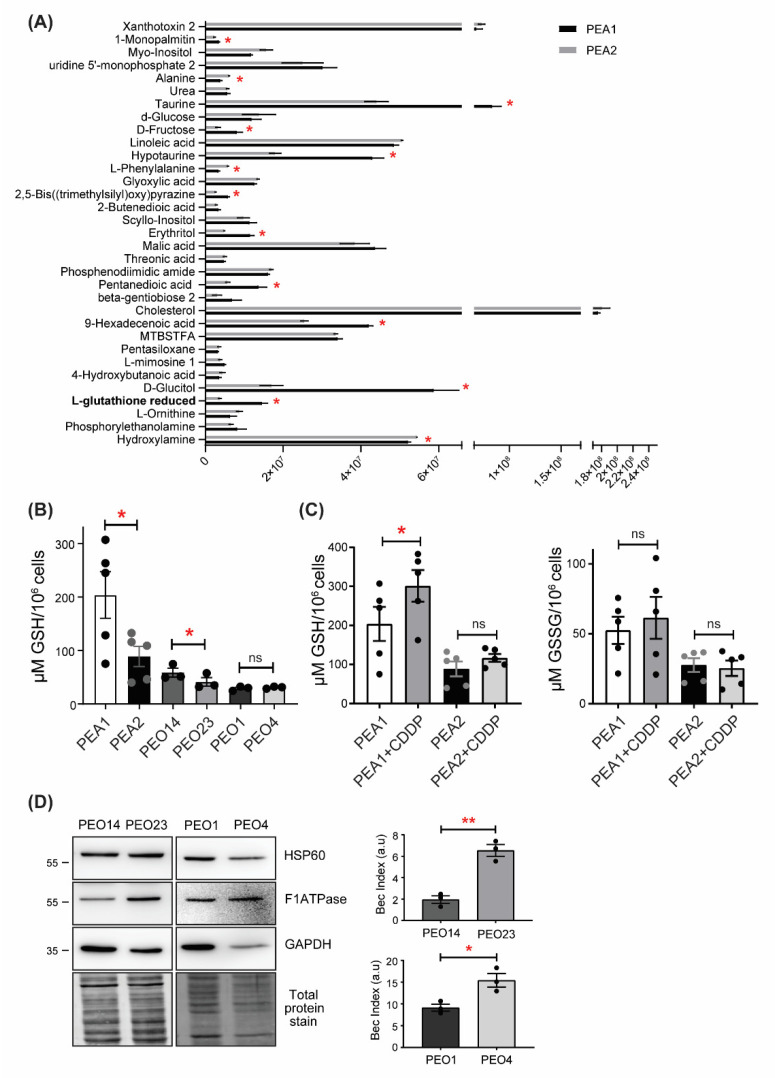
Cisplatin-resistant high grade serous ovarian cancer (HGSOC) cells show decreased GSH levels. (**A**) Untargeted metabolomics analyses of the drug sensitive and resistant PEA1/PEA2 cell lines. Data are expressed as mean ± S.E.M (*n* = 3). Asterisks above the bars represent statistically significant differences (*p*-value < 0.05) calculated using the Student’s *t*-test. (**B**) Determination of GSH levels in cisplatin-sensitive and -resistant OC cells under basal conditions. (**C**) Determination of GSH and GSSG levels in PEA1 and PEA2 cells treated with 20 μM and 40 μM of cisplatin, respectively, for 30 min. GSH and GSSG levels were measured by a colorimetric assay as outlined in the Materials and Methods. Data are expressed as mean ± S.E.M. from three independent experiments, each with technical duplicates. Significance was assessed by paired Student’s *t*-test (* *p*-value < 0.05, ** *p*-value < 0.01, ns *p*-value > 0.05). (**D**) BEC index analysis on PEO1/PEO4 and PEO14/PEO23 cells. Cells were harvested and total protein lysates were immunoblotted with anti-F1ATPase, anti-HSP60, and anti-GAPDH antibodies. Immunoreactive bands were quantified by using ImageJ and BEC index was calculated by the formula F1ATPase/HSP60/GAPDH (see Section 2 Materials and Methods for details). Data are expressed as mean ± S.E.M. from three independent experiments. Significance was assessed by two-tailed Student’s *t*-test (* *p*-value < 0.05, ** *p*-value < 0.01).

**Figure 2 antioxidants-11-01544-f002:**
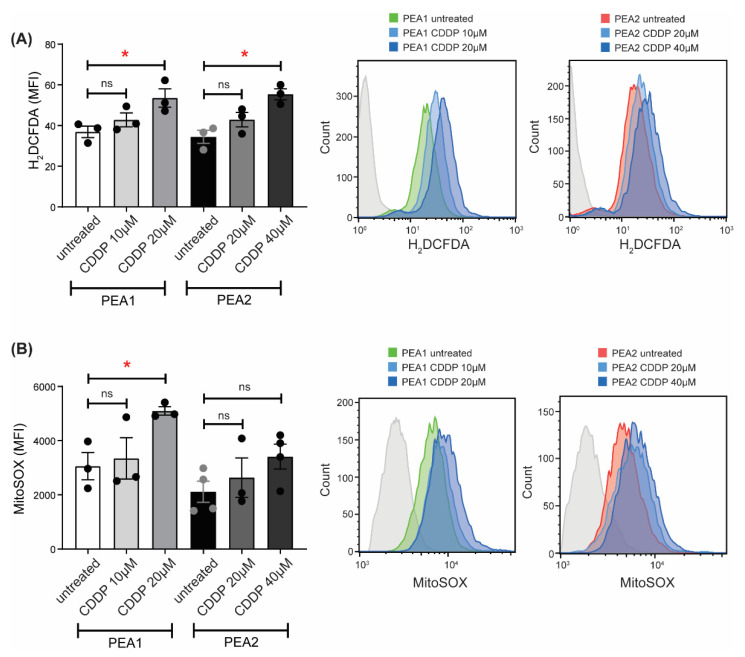
Cisplatin increases intracellular reactive oxygen species (ROS) levels. Cytosolic ROS (**A**) and mitochondrial superoxide (**B**) levels in PEA1 and PEA2 cells treated with cisplatin at indicated concentrations for 24 h were quantified by flow cytometry using H_2_DCFDA dye and MitoSOX, respectively. Data are represented as mean fluorescence intensity and expressed as mean ± S.E.M. from three independent experiments. Significance was assessed by two-tailed Student’s *t*-test (* *p*-value < 0.05, ns *p*-value > 0.05). Representative histograms of H_2_DCFDA and MitoSOX expression are shown in the overlay.

**Figure 3 antioxidants-11-01544-f003:**
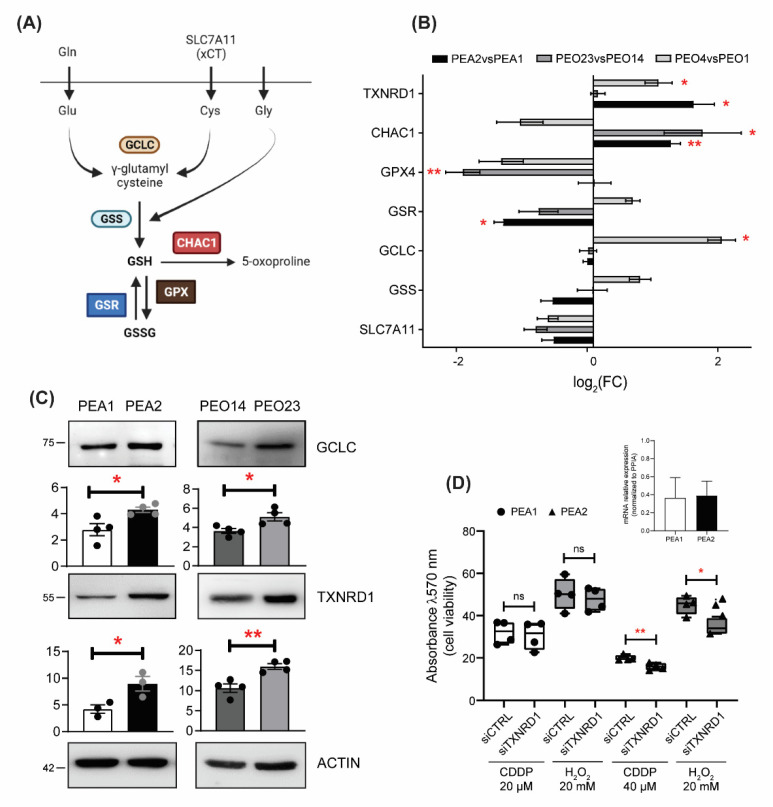
Drug-resistant HGSOC cells exhibit a remodeling of antioxidant systems. (**A**) Schematic representation of glutathione synthesis, recycling, and degradation. (**B**) Real-time RT-PCR analysis of indicated genes in platinum-resistant PEA1, PEO23, and PEO4 cells compared to their matched-sensitive counterparts PEA1, PEO14, and PEO1 cells. Data are expressed as mean ± S.E.M. of −ΔΔCt (log2FC) from four independent experiments with technical triplicates each. Numbers indicate the statistical significance (*p*-value), based on the Student’s *t*-tests performed on ΔCt values (significant values highlighted in red). (**C**) Total lysates obtained from cisplatin-sensitive and -resistant OC cells were separated by SDS-PAGE and immunoblotted with the indicated antibodies. Images are representative of three independent experiments. Bar graphs below each image represent densitometric quantification of bands, expressed as mean ± SEM (*n* = 3). Significance was assessed by Student’s *t*-test (* *p*-value < 0.05, ** *p*-value < 0.01). (**D**) Viability assays performed in PEA1 and PEA2 cells following TXNRD1 silencing and treatment with either cisplatin (20 µM and 40 µM, respectively) for 48h or H_2_O_2_ (20 mM) for 30 min. Data are expressed as mean ± SEM of four independent experiments, with technical triplicates each. Significance was assessed by Student’s *t*-test (* *p*-value < 0.05, ** *p*-value < 0.01, ns *p*-value > 0.05). The insert shows the degree of TXNRD1 silencing expressed as fold change compared to the respective control-directed siRNA transfected cells (*n* = 2).

**Figure 4 antioxidants-11-01544-f004:**
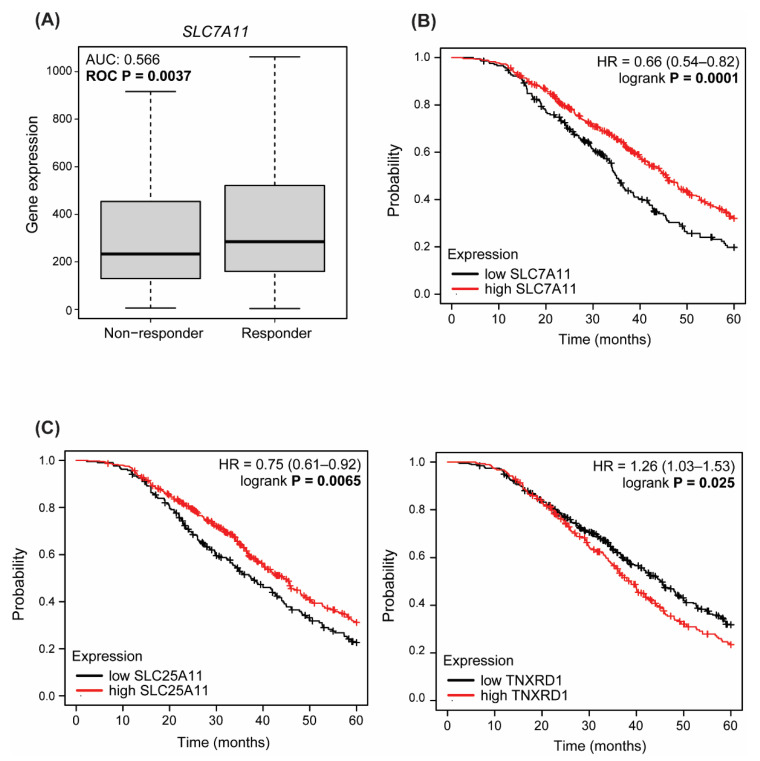
Antioxidant enzymes correlate with response to therapy and survival in HGSOC patients. (**A**) Expression levels of SLC7A11 in HGSOC patients, grouped for their pathological response to platinum-based therapy (complete response vs. residual disease after completing the therapy). Statistical significance of differences in gene expression levels in responders (*n* = 432) and non-responders (*n* = 202) was evaluated by the Mann–Whitney test. (**B**,**C**) Kaplan–Meier estimates of survival obtained on the Kaplan—Meier plotter analyzing post-progression survival in HGSOC patients treated with platinum-based chemotherapy.

**Figure 5 antioxidants-11-01544-f005:**
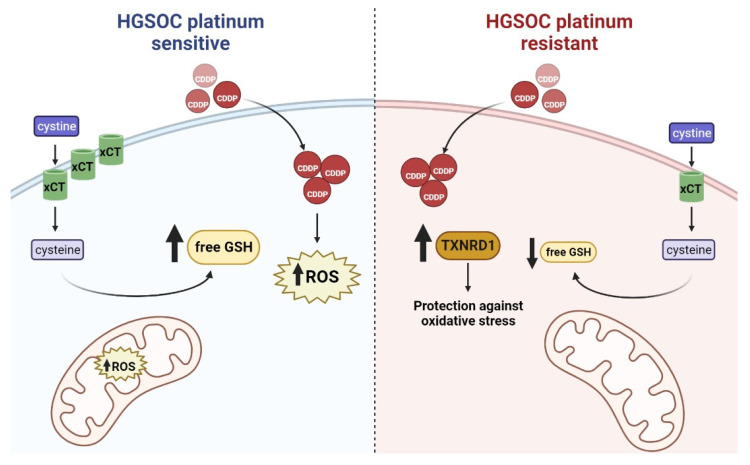
Platinum sensitive HGSOC cells (**left**) display higher levels of cytoplasmic GSH, which has been proposed as the main cytoplasmic target of cisplatin (CDDP) that can enter the cell through passive diffusion via the copper transporter. CDDP treatment causes accumulation of ROS in both the cytosolic compartment and the mitochondria, in which GSH is normally imported through the oxoglutarate carrier (OGC)/SLC25A11. Platinum resistant clones (**right**) evade this pathway by downregulating xCT and/or the enzymes involved in GSH synthesis and recycling. GSH levels decrease and, as a compensatory mechanism, platinum resistant HGSOC cells have higher thioredoxin reductase (TXNRD1).

**Table 1 antioxidants-11-01544-t001:** Primers used for quantitative polymerase chain reaction (qPCR) analyses.

Target Genes (Accession n.)	Forward (5′-3′)	Reverse (5′-3′)
PPIA (NM_021130.5)	CTGCACTGCCAAGACTGA	GCCATTCCTGGACCCAAA
GSS (NM_000178.4)	GCGGAGGAAAGGCGAACTA	AGAGCGTGAATGGGGCATAG
GCLC (NM_001498.4)	ACGGAGGAACAATGTCCGAG	TCCACTGGGTTGGGTTTGAC
GSR (NM_000637.5)	GTGGAGGTGCTGAAGTTCTCC	AACCATGCTGACTTCCAAGC
GPX4 (NM_002085.5)	CCTGGACAAGTACCGGGGC	CTTCGTTACTCCCTGGCTCCTG
xCT (NM_014331.4)	TGAAATCCCTGAACTTGCGAT	TCTGGATCCGGGCGCT
TXNRD1 (NM_182729.3)	CGATCTGCCCGTTGTGTTTG	TATTGGGCTGCCTCCTTAGC
CHAC1 (NM_024111.6)	TTCTGGCAGGGAGACACCTT	GCCTCTCGCACATTCAGGTA
SLC25A11 (NM_003562.5)	CGTCAAGTTCCTGTTTGGGG	AGTGTAAATGCCCCTCAGGC

PPIA: cyclophilin A; GSS: glutathione synthetase; GCLC: glutamate-cysteine ligase catalytic subunit; GSR: glutathione-disulfide reductase; GPX4: glutathione peroxidase 4; xCT: solute carrier family 7 member 11 (SLC7A11); TXNRD1: thioredoxin reductase 1; CHAC1: ChaC glutathione specific gamma-glutamylcyclotransferase 1; SLC25A11: solute carrier family 25 member 11.

## Data Availability

Data is contained within the article and Appendix A.

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
