# Peer review of "Decreased Levels of GSH Are Associated with Platinum Resistance in High-Grade Serous Ovarian Cancer"

_antioxidants, 2022, doi:10.3390/antiox11081544_

Round 1
Reviewer 1 Report
Summary: The article is interesting and asks questions about reduced glutathione and the GSH pathway in platinum resistance, and important topic that needs addressing in ovarian cancer. Using cell lines from patients pre-and post platinum resistance it was found that reduced glutathione was lower in resistant lines, and GSH was reduced in response to cisplatin.
Minor comments:
In the discussion, comment whether the resistant cells could be re-sensitized to platinum by adding exogenous GSH.
I think you should include the following reference as well as or instead of the current #32: Fekete, J. T.; Osz, A.; Pete, I.; Nagy, G. R.; Vereczkey, I.; Gyorffy, B., Predictive biomarkers of platinum and taxane resistance using the transcriptomic data of 1,816 ovarian cancer patients. Gynecol Oncol. 2020 Mar;156(3):654-661. doi:10.1016/j.ygyno.2020.01.006.
Reviewer 2 Report
The manuscript “Decreased levels of GSH are associated with platinum resistance in high grade serous ovarian cancer” is a research article that investigated the correlation of GSH levels to the sensitivity to cisplatin in HGSOC. I really appreciate the work performed by authors; the manuscript is well written and the conclusions are supported by the data. However, there are some points that authors should address in order to consider the manuscript suitable for publication.
1. In the affiliations is written “e-mail@e-mail.com” ; please fix it.
2. Line 39: please report the incidence in percentage of HGSOC respect to all ovarian cancers.
3. Line 75: authors correctly state that when the levels of ROS overcome the antioxidant defenses, the following detrimental oxidative stress can lead to cell death. However, they should underline that following this statement, increasing intracellular oxidative stress, utilizing natural or synthetic compounds, has been proposed as a possible therapeutical strategy to treat cancer.
4. Line 147: how was the cisplatin concentration 20 and 40uM chosen?
5. Line 153: I think authors mean “ThermoFisher”
6. Line 193: please specify the blocking solution used and the time and temperature of incubation of primaries and secondaries antibodies.
7. In all figures please change “uM” with “µM”.
Author Response
Please see the attachment.

This manuscript is a resubmission of an earlier submission. The following is a list of the peer review reports and author responses from that submission.
Round 1
Reviewer 1 Report
Overall, the authors present convincing evidence that GSH levels are reduced in chemoresistant isogenic cell lines and that components of the GSH synthesis pathway correlate with post-progression survival. The majority of the study relies on in vitro experiments and is somewhat descriptive. Despite these deficiencies, the overall findings are of interest.
Minor Comments:
1) The authors should indicate how the doses of cisplatin were chosen in Fig 2.
2) The authors should provide more detail on the Bec index and how well this correlates with more traditional measures of oxidative phosphorylation such as the Seahorse assay.
Reviewer 2 Report
The authors started this project with a finding from the metabolomics showing that PEA2 cells display reduced GSH levels compared to PEA1 cells, and then attempted to test a hypothesis that cisplatin treatment could positively select cancer cells which are independent from GSH for the maintenance of redox balance, and thus less sensitive to cisplatin-induced oxidative stress. However, the data presented in this manuscript are not sufficient to support this hypothesis. Although three pairs of sensitive-resistant cell lines were tested, the results are not always consistent. The genes analyzed in this project are not necessarily related to oxidative stress and GSH metabolism.
Other concerns:
1. If the authors hypothesized that a decrease of GSH in chemoresistant cells could result from cisplatin binding of GSH, treatment with cisplatin should decrease the GSH level in PEA1 cells but not increase GSH as shown in Fig. 1C.
2. PEA2 cells need a doubled amount of cisplatin to reach the same ROS increase observed in PEA1 cells in Fig. 2A, could also be due to the reduced cisplatin uptake and/or increased cisplatin efflux.
3. Fig. 3D, data are presented as fold change, but it is unclear how the fold change was calculated.
4. Literature 34 does not show that PTGS2, CHAC1 and xCT are robust markers of oxidative cell death. Thus, using these gene expression to reflect oxidative cell death is not appropriate.
5. All genes analyzed in Fig. 4 are not specific antioxidant enzymes. The survival curves shown in Fig. 4 do not indicate a correlation between expression of these genes and acquisition of platinum resistance.
6. Expression level of SLC7A11 is not necessarily correlated with GSH production and cisplatin resistance.
Reviewer 3 Report
In this manuscript authors found decreased GSH levels in resistant HGSOC cell lines. Moreover, they found reduced levels of enzymes involved in synthesis and recycling of GSH with a compensatory increased expression of thioredoxin reductase. Cisplatin treatment increased GSH. Furthermore, expression of the cysteine-glutamate antiporter xCT was directly correlated with post-progression survival of HGSOC patients and was reduced in patients with cisplatin chemoresistance.
This study is very interesting but presents some flaws. In particular:
- Introduction: Although authors properly introduced the role of GSH in cisplatin resistance onset in ovarian cancer, they did not mention the NRF2/KEAP1 pathway that play a pivotal role in cisplatin resistance onset in this disease. This pathway deserves to be discussed because it is involved in the regulation of GSH levels (but also many antioxidant enzymes including GCLC and Thioredoxins) in cisplatin resistance ovarian cancer cell lines (including PEO1 and PEO4) since these cells show increased GSH levels and increased NRF2 expression (PMID: 35453348). Moreover, NRF2/KEAP1 pathway can modulate GCLC and Thioredoxins further supporting the results obtained by the authors.
-Line 405: authors should underline that their results are in accordance to what is reported in literature by other studies in which targeting antioxidant enzymes resulted in increased sensitivity of cancer cells to cisplatin (PMID: 32093309). Indeed, suppressing the antioxidant defenses, including reducing GSH content, is a promising strategy for cancer treatment (PMID: 35453297). All these considerations should be reported in the discussion section, since they strongly support the reliability of the findings presented by authors.
-Western Blotting analyses: Primary and secondary antibody dilutions must be reported
- Figure 4B: Please modify the format of logrank P according to the rest of the figure
- Lines 335-337 and Figure 4A: Authors stated that SLC7A11 (xCT) expression was lower in patients who do not respond to platinum-based therapy but figure 4A does not report any significance or p value
An accurate revision of typing errors is recommended
Round 2
Reviewer 2 Report
Although the authors have addressed some concerns, the major one "the data presented in this manuscript are not sufficient to support this hypothesis" was not appropriately addressed. There is no direct evidence to show "decreased levels of GSH are associated with platinum resistance". The association between antioxidant enzyme expression and HGSOC patient survival in Fig. 4 is not a direct evidence to show the relationship between GSH and cisplatin resistance. In addition, the experiments were not logically designed.
Reviewer 3 Report
manuscript has been significantly improved and can be accepted in the present form.